# Revealing a Novel Antigen Repressor of Differentiation Kinase 2 for Diagnosis of Human Visceral Leishmaniasis in India

**DOI:** 10.3390/pathogens11020120

**Published:** 2022-01-20

**Authors:** Anirban Bhattacharyya, Mohd Kamran, Sarfaraz Ahmad Ejazi, Sonali Das, Nicky Didwania, Rahul Bhattacharjee, Mehebubar Rahaman, Rama Prosad Goswami, Krishna Pandey, Vidya Nand Ravi Das, Pradeep Das, Saswati Gayen, Nahid Ali

**Affiliations:** 1Infectious Diseases and Immunology Division, Indian Institute of Chemical Biology, Kolkata 700032, India; anirbanbiomedical@gmail.com (A.B.); mohdkamran9808@gmail.com (M.K.); sarfaraz.ejazi@hotmail.com (S.A.E.); dassonali571@gmail.com (S.D.); didwanianicky@gmail.com (N.D.); rahuljonty0798@gmail.com (R.B.); 2Department of Tropical Medicine, School of Tropical Medicine, Kolkata 700073, India; rmehbub@gmail.com (M.R.); drrpgoswami@gmail.com (R.P.G.); 3Department of Molecular Biology, Rajendra Memorial Research Institute of Medical Sciences, Patna 800007, India; drkrishnapandey@yahoo.com (K.P.); dasvnr@icmr.org.in (V.N.R.D.); drpradeep.das@gmail.com (P.D.); 4Department of Microbiology, VijaygarhJyotish Ray College, Bejoygarh 700032, India

**Keywords:** Leishmaniasis, diagnosis, antigen, ELISA, immunoblotting

## Abstract

Visceral leishmaniasis (VL) is one of the major global health concerns due to its association with morbidity and mortality. All available diagnostic tools have been, until now, unable to provide a very specific and cost-effective mode of detection for VL globally. Therefore, the design of robust, specific, and commercially translatable diagnostic tests is urgently required. Currently, we are attempting to identify and explore the diagnostic potential of a novel parasite antigen. Repressor of differentiation kinase 2 (RDK2), a serine/threonine kinase, has a versatile role in parasite life cycle progression. However, its role as a diagnostic candidate for VL has not been investigated. Herein, we cloned and over-expressed LdRDK2 and studied the recombinant RDK2 for the diagnosis of human VL using serum and urine samples. In silico analysis predicted that RDK2 is conserved among *Leishmania* species with the least conservation in humans. RDK2 developed immune-reactive bands with antibodies present in VL patients’ sera, and it demonstrated no cross-reactivity with sera from healthy controls and other diseases. Additionally, RDK2 antigen demonstrated a significant reactivity with IgG antibodies of VL patients’ sera, with 78% sensitivity and 86.67% specificity as compared to healthy controls and other diseases. Furthermore, we evaluated its utility for non-invasive diagnosis of VL using patients’ urine samples and found 93.8% sensitivity and 85.7% specificity. RDK2 was found to have better sensitivity and treatment response in patients’ urine compared to serum samples, indicating its role as a promising point of care (POC) antigen. In a nutshell, we explored the role of RDK2 as a potential diagnostic marker for VL in both invasive and non-invasive modes as well as its utility as a promising POC antigen for treatment response cases.

## 1. Introduction

Visceral leishmaniasis (VL), or kala-azar, caused by *Leishmania donovani*, is one of the deadliest parasitic diseases in terms of outbreak and mortality. Historically, the disease has been endemic mainly in the Indian Subcontinent, Latin America, and East Africa [1]. According to the World Health Organization (WHO) 2019 report, 50,000 to 90,000 new VL cases occur annually, of which 90% of cases arise from Brazil, Ethiopia, Eritrea, India, Iraq, Kenya, Nepal, Somalia, South Sudan, and Sudan (https://www.who.int/news-room/fact-sheets/detail/leishmaniasis, accessed on 20 May 2021). The onset of VL causes symptoms such as irregular bouts of fever, weight loss, splenomegaly, hepatomegaly, and anaemia that can overlap with symptoms of other diseases such as malaria and dengue [2,3].

Laboratory diagnosis of this disease can be performed using several different approaches, such as microscopic examination of tissue aspirates, enzyme-linked immunosorbent assay (ELISA), indirect immunofluorescence (IIF), indirect hemagglutination (IHA), and direct agglutination test (DAT) [4,5,6,7,8]. Amongst these, direct observation of amastigotes in Geimsa stained tissue aspirates is preferred as the gold standard diagnosis technique. However, the main drawback to this process stems from the painful and complex methodology of sample collection from bone marrow and splenic aspirates, which requires considerable levels of skill and hospital care facilities [9]. Other techniques such as ELISA, IIF, IHA, and DAT have been popular for decades. However, these assays have their own limitations, including consumption of time, performance variability, and the need for multiple sample tests. In recent years, development of molecular tests such as quantitative real-time polymerase chain reaction (qPCR) and loop-mediated isothermal amplification (LAMP) are being performed in order to achieve high specificity [10,11,12]. However, these require expensive equipment and lab accessories that result in a high cost of diagnosis for the infected people suffering from malnutrition, population displacement, poor housing, a weak immune system, and a lack of financial resources [13].

Currently, rapid diagnostic tests (RDTs) such as lateral flow assays (LFA), particle agglutination, immunodot, and immunofiltration are emerging worldwide for rapid diagnosis of VL. Amongst these RDTs, the rK39-based RDT is, at present, the most potent diagnostic tool for endemic regions in the Indian Subcontinent, but rK39 fails to diagnose Latin American and East African cases, especially due to lower sensitivities [14,15,16]. Further, false positive results of rK39-based RDT (39.0%) and DAT (53.0%) in 15 years after treatment of VL patients in the endemic region of Muzaffarpur, India, indicated that rK39 failed to differentiate among active infection, previous infection, or previous asymptomatic infection [4,12]. Therefore, these limitations hinder the path of using rK39-based RDT as a point-of-care (POC). Interestingly, our group has previously explored the diagnostic potential of *leishmania* membrane antigens (LAg) that can detect the presence of infection specific antibodies in the sera and urine of active VL patients with high specificity and sensitivity [16,17,18,19]. Furthermore, with no false positivity in urine samples after treatment of VL patients, LAg-based RDT proved to be an applicant for POC [6]. However, the major limitation associated with the usage of purified antigens is batch-to-batch variation in production from different passages of *Leishmania* culture. Additionally, the expensive method for large scale production of LAg would further enhance the cost of the kit [20]. Therefore, development of recombinant antigen based, inexpensive, non-invasive diagnostic toolkits is the need of the hour, because it will enhance the accuracy of VL diagnosis globally and it can become a POC [21].

In the last few years, protein kinases (PKs) of *Leishmania* sp., are getting highlighted as important cell signalling molecules that bear the potential of being therapeutic as well as diagnostic markers [9,22]. The recombinant antigen repressor of differentiation kinase 2 (RDK2) is a serine/threonine kinase that has a role in eukaryotic cell-cycle regulation and belongs to the NIMA-related kinases (*NEK PK)* family. Previously, the roles of RDK2 were studied in controlling the differentiation of *Trypanosoma brucei* from short stumpy bloodstream form (BSF) to insect procyclic form (PCF) [22]. Besides *T.*
*brucei*, these mitotic kinases are also highly conserved in *Leishmania sp.*, and they have vital involvement in the regulation of the cell cycle. These findings lend hope that specific diagnostic approaches may be achieved using these mitotic kinases such as RDK2 from *L. donovani.* In the current study, we have evaluated the diagnostic potential of the recombinant RDK2 antigen from *L. donovani* through ELISA and immunoblot assays using archived VL patients’ sera and urine samples.

## 2. Materials and Methods

### 2.1. Bioinformatics Analysis of RDK2 Gene

*L. donovani* RDK2 gene sequence was obtained from theTriTrypDBdatabase version 3.2 (http://tritrypdb.org/tritrypdb/, accessed on 16 January 2017). BLAST searches were performed via the NCBI (http://blast.ncbi.nlm.nih.gov/Blast.cgi, accessed on 16 January 2017). Multiple sequence alignment was performed using Jalview software version 2.1 using amino acid sequences of RDK2 from *L. donovani, L. infantum, L. mexicana, L. major, T. brucei, T. cruzi**,*
*and H. sapiens.*

### 2.2. Biological Samples

A total of 88 archived human sera and 105 urine samples were collected from the School of Tropical Medicine (STM) in Kolkata, and the Rajendra Memorial Research Institute of Medical Sciences (RMRIMS) in Patna, India, for the study. All active VL patients were parasitologically confirmed based on commercially available rK39-RDT (InBios Int. Inc., USA) tests. Among the serum samples, 50 were from active VL patients (AVL), 5 from endemic healthy controls (EHC), and 10 from non-endemic healthy controls (NEHC). In addition, 10 serum samples from other diseases (OD) symptomatically similar to VL were also collected, and these included 3 each from malaria and viral fever, and 2 each from typhoid and tuberculosis. Samples from EHC, NEHC, and OD were used as negative controls. All of the samples were kept frozen at −20 °C until they were used. Among the urine samples, 49 were from active VL patients (AVL), 10 EHC, 21 NEHC, 6 FU, and 19 OD which includes 5 each from malaria, tuberculosis, and viral fever, and 4 samples from typhoid patients served as negative controls. Urine samples were preserved by adding 0.1% sodium azide and stored at 4 °C until use. Follow-up (FU) serum, total of 13, and urine, total of 6, were also collected at approximately six months post-treatment.

### 2.3. Ethics Statement

All of the human samples used in this study were ethically approved by the Ethical Committee on Human Subjects of CSIR- Indian Institute of Chemical Biology, Kolkata (No. IICB/IRB/2021/I), School of Tropical Medicine, Kolkata (IEC Ref. No. CREC-STM/2020-AG-11), and Rajendra Memorial Research Institute of Medical Sciences, Patna (RMRIMS/IEC/18.07.2014). Written informed consents were also obtained individually from each patient and healthy volunteer who participated in the study.

### 2.4. Cloning, Overexpression and Purification of L. donovani RDK2

The cloning of the RDK2 gene from *L. donovani* strain AG83 (ATCC PRA-413) was carried out in the pET28a vector and then expressed in the Rosetta (DE3)strain of *Escherichia coli* cells and produced N-terminal His-tagged RDK2 protein. Genomic DNA from *L. donovani* promastigotes was isolated and the RDK2 gene PCR-amplified and cloned into the pET28a vector using NcoI/HindIII restriction sites. At 37 °C, RDK2 was overexpressed in 1000 mL of Luria–Bertaini media in the presence of 0.5 mM isopropyl-D-1-thiogalactopyranoside (IPTG) with an optical density (OD) of about 0.4 to 0.6. The RDK2 protein was purified through Ni-NTA column chromatography using a 1000 mL culture of *E. coli*. The cultures were then harvested after 4 h, and the cell pellets were suspended in 10 mL of bacterial lysis buffer (25 mM Tris-HCl, 300 mM NaCl, 1 mg/mL of lysozyme (Roche), and 1 mM phenylmethylsulfonyl fluoride (PMSF)(pH 8.0)) followed by sonication using an ultrasonicator (Misonix, Farmingdale, NY, USA). Then, the lysates (10 mL) were centrifuged at 12,000 rpm for 30 min and inclusion bodies in the pellets were subjected to purification under denaturing conditions. Purification of urea denatured eluted fractions was subjected to dialysis with a gradually decreasing concentration of urea to provide ensure folding of the proteins. These protein fractions were further concentrated using Amicon Ultra centrifugal filter devices (Millipore Corporation, Burlington, MA, USA). The purification of proteins was confirmed by performing a 12% SDS-PAGE from purified fractions (20 µL/lane) followed by Coomassie brilliant blue staining. The amount of protein in each case was estimated by the method of Lowry [23].

### 2.5. Enzyme-Linked Immunosorbent Assay (ELISA)

The performance of the recombinant RDK2 was evaluated by ELISA, and the levels of IgG antibodies (specific for antigen) were detected by enzyme-linked immunosorbent assay (ELISA). The experiment was performed by coating recombinant RDK2 antigen (1 mg/mL) in 96-well ELISA plates (Nunc) overnight at 4 °C diluted in phosphate buffer (PB). The next day, the wells were blocked with 1% bovine serum albumin (BSA) (200 µL/well) at 37 °C for 1 h. For primary antibody binding, serum (100 µL/well) and urine samples (100 µL/well) were added at a dilution of 1:2000 [19] and 1:10 [16], respectively, and incubated at 37 °C for 2 h, followed by secondary antibodies at a 1:3000 dilution of horseradish peroxidase (HRP)-conjugated anti-human IgG. Finally, the presence of bound IgG was detected by adding 3,3’,5,5’-Tetramethylbenzidine (TMB) (Sigma-Aldrich, St. Louis, MI, USA), as the substrate, and the reaction was stopped by the addition of 2N H_2_SO_4_. Optical density values were obtained at 450 nm using a microplate spectrophotometer (Thermo Fisher Scientific, Waltham, MA, USA).

### 2.6. Immunoblot Assay

Immunoblot assay or Western blot against recombinant RDK2 antigen (20 µg/lane) was carried out using 12% SDS-PAGE. The resolved proteins were electrophoretically transferred onto the nitrocellulose using a transblot apparatus (Bio-Rad Laboratories, Hercules, CA, USA) at 1.5 A constant current for 15 min. After confirming transfer of the protein onto membranes with Ponceau S staining, each lane was cut into strips and was then blocked with 5% BSA in Tris-buffered saline (TBS) for 60 min and incubated with primary antibodies present in serum at 1:2000 dilutions for overnight at 4 °C with constant shaking. The next day, secondary antibodies, HRP-conjugated goat anti-human IgG antibody at 1:3000 dilutions, were added and incubated for 1hr at room temperature with constant shaking. Strips were washed 5 times with a wash buffer at a 5 min interval, and then chemiluminescent HRP substrate (Sigma-Aldrich) was added to each strip (blots). Images were developed using the Gel Doc system (Bio-Rad, Hercules, CA, USA) and analysed in Image Lab software (version 54.2.1, Bio-Rad).

### 2.7. Data Analysis

Statistical analysis was performed using the GraphPad Prism 8.0 software. To determine the sensitivity and specificity of ELISA cutoff values, the Receiver Operating Curve (ROC) at 95% confidence intervals was used. Furthermore, to calculate the statistical significance, the Mann–Whitney U test was performed, and differences were considered statistically significant when the *p* value was <0.05.

## 3. Results

### 3.1. Identification, Sequence Analysis, Cloning and Purification of RDK2

A complete open reading frame (ORF) of the *L. donovani* RDK2 (LdBPK_313070.1) gene (1326 bp) has been retrieved from the triTrypDB database. The gene encodes a 441 amino acid long RDK2 protein with an approximate molecular weight of 49 kDa. The homology and pan-species conservation of RDK2 have been analysed by comparing amino acid sequences of RDK2 obtained from *L. infantum, L. mexicana, L. major, T. brucei, T. cruzi*, and *H. sapiens*. The sequences were aligned using a multiple sequence alignment tool (MSA). According to the conservation peaks obtained from the MSA tool, RDK2 is conserved throughout *Leishmania* and *Trypanosomes* with the least conservation in humans (Figure 1). 

In order to obtain recombinant and purified RDK2, the 1326 bp long ORF of the RDK2 gene has been PCR amplified from *L. donovani* genomic DNA and cloned in the pet28-kan+ vector (Figure 2A). Using 0.5 mM IPTG mediated induction, the recombinant RDK2 has been overexpressed in the Rosetta strain of *E.coli* (Figure 2B). Recombinant RDK2 was purified from induced bacterial inclusion bodies using a urea-based solubilization method. SDS-PAGE analysis determined the purity and homogeneity of recombinant RDK2, which corresponded to 49 kDa (Figure 2C).

### 3.2. ELISA Test with RDK2 for Detection of Serum IgG Antibodies

To assess the recognition potential of recombinant RDK2 for VL diagnosis, the reactivity of this protein in ELISA was carried out with the serum of VL positive patients as well as controls that include EHC, NEHC and OD as mentioned above in the Materials and Methods section. With 50 VL positive serum samples and 38 samples from control groups comprising both endemic (n = 5) and non-endemic healthy individuals (n = 10), along with patients having other diseases (n = 10) and follow-up (FU, n = 13), the presence of antigen-specific IgG antibodies was determined through ELISA and presented as optical density values. A cutoff value (0.44) was determined from the ROC curve where maximum sensitivity and specificity were observed. A Sensitivity of 78% of the confirmed VL cases was detected through serum RDK2 ELISA and a specificity of 86.67% was observed (Figure 3A). Moreover, antibody levels of VL patients were found to be statistically significant in comparison to endemic healthy controls and other diseases. The ROC curves obtained for the antigen RDK2 is shown in Figure 3B.

### 3.3. ELISA Test with RDK2 for Detection of Urine IgG Antibodies

We further investigated the reactivity of antigen-specific antibodies against recombinant proteins in the urine samples of VL positive patients. For this, 49 samples from VL positive patients, 21 samples from non-endemic healthy controls (NEHC), 10 samples from endemic healthy (HC) controls, and 19 samples from other diseases (OD) were obtained. ROC curves were used to determine the cutoff values (0.29), which reflect the performance of the antigen. ELISA analysis showed a sensitivity of 93.8% and a specificity of 85.7% (Figure 4A) with RDK2 urine samples. In the case of other diseases, few samples were above the cutoff line, depicting very little cross-reactivity. The ROC curves obtained for RDK2 are shown in Figure 4B.

In order to assess the treatment response, follow-up samples containing 13 sera and 6 urine samples of *Leishmania* infected patients, (post 6-month treatment with AmBisome), were collected. With the exception of one sample, the levels of disease-specific IgG antibodies in ELISA analysis of urine sample were lower than the cutoff values (Figure 4A), while with serum, most of the samples were above the cutoff line (Figure 3A). Altogether, these independent results obtained from serum and urine samples show that RDK2 is almost equally sensitive and specific, but it could be used as a better prognostic marker for non-invasive diagnosis of VL using urine.

### 3.4. Immunoblot Assay of RDK2 with Serum Samples

After evaluating the diagnostic and prognostic potential of recombinant antigen RDK2 in serum and urine ELISA, we screened the antigen’s reactivity in the immunoblot assay for confirmation of the sero-diagnosis of VL. The antigen’s reactivity with multiple samples was evaluated in two separate immunoblot assays. In the first assay, nine active VL patients’ serum samples (VL1 to 9) were used, and in another four samples of controls, two each of non-endemic healthy controls (HC1, HC2), endemic healthy controls (EC1, EC2), and three samples of other similar diseases, malaria (Ma), viral fever (Vi), and tuberculosis (Tb) for diagnosis, and two active VL patient’s sera samples (VL1 and VL2) (Figure 5). The RDK2 showed positive reactivity with all VL sera in the immunoblot assay (Figure 5A) and was negative with endemic, non-endemic healthy individuals, and other diseases (Figure 5B).

## 4. Discussion

An early, affordable, and correct diagnosis is a must for providing suitable treatment and effective control of *Leishmania* transmission. However, the challenges associated with the VL control revolve around improving the performance of diagnostic tests for human use over a long period. Since the disease is still present in many parts of the world, the search for a better candidate for VL diagnosis remains. In order to capture the patients’ antibodies through immunological tools, several recombinant antigens have been developed for ELISA and immunoblot assays. The diagnostic potential of an immunological assay is largely associated with the reactivity of the antigens used. In this context, information about the sensitivity and specificity of the antigen involved in the assay is necessary to allow an accurate diagnosis. In the current study, we have generated a recombinant version of the antigen RDK2 from *L. donovani* and evaluated its performance in diagnosis as well as in treatment response, utilizing serum and urine samples of Indian VL patients.

A large number of immunological assays such as DAT, ELISA, and immunochromatographic tests have been developed in recent years, varying vastly in their ability to diagnose VL. DAT represents an antigen-antibody agglutination-based test that uses whole promastigote antigens for diagnosis. DAT has been evaluated in many VL endemic regions, mainly in the African subcontinent. In VL diagnosis, DAT gives sensitivity in the range of 70.5–100% and specificity of 53–100% [24,25]. However, the need for serial sample dilutions, lengthy incubation, and variations in the quality of antigen render this technique outdated for field diagnosis. Employment of leishmanial antigens in the ELISA method has spurred the diagnostic studies promptly. Earlier, in our previous reports, we have purified *L. donovani* native membrane antigens, (LAg), and evaluated them in ELISA. The diagnostic potential of LAg demonstrated 100% and 97.94% sensitivity towards serum and urine samples, respectively [16,19]. Further, different antigenic components of LAg, such as 31, 34, 36, 45, 51, 63, 72, 91, and 97 kDa, were electroeluted and evaluated separately in ELISA. A 100% sensitivity was achieved by 34 and 51 kDa purified antigens [2,26].

Variations in the isolation procedure of native antigens may result in inconsistency in antigen performance. Introduction of recombinant antigens in diagnosis appears to be the best alternative to obtain a specific antibody response in the host. Recombinant antigens do not require parasite culturing and they allow large-scale production with remarkable purity at a minimum cost. Several defined recombinant proteins have been characterized and evaluated for use in *Leishmania* diagnosis. Out of these antigens, rK39 is at the top for VL diagnosis with 67% to 100% sensitivity and 93% to 100% specificity. However, the drawback associated with this antigen is its low sensitivity in the African region, false positivity (15% to 32%) in endemic healthy persons in the Indian subcontinent, and cross-reactivity with symptomatic diseases such as malaria and enteric fever [27]. We, in our previous studies, have evaluated the recombinant proteins such as cysteine protease C (CPC), glycoprotein 63 (GP63), and elongation factor 1-α (EF1- α) in indirect ELISA. The sensitivity obtained with Indian sera was 98.15%, 92.59% and 96.29% for CPC, GP63, and EF1- α, respectively. With urine samples, we observed 96%, 90%, and 84% sensitivities for the respective antigens [28]. Another recombinant antigen, *L. donovani*–otubain cysteine peptidase (Ld-OCP) in serological diagnosis, demonstrated 96.92% and 97.43% sensitivity and specificity in diagnosing VL [21].

Our previous reports have focused on the development of precise VL diagnosis by replacing native leishmanial antigens with comparatively better recombinant antigens. In this context, a novel recombinant kinase antigen, RDK2, was identified and validated in *Leishmania* diagnosis. The involvement of protein kinases has proved to be potentially viable as diagnostic markers as well as drug targets for the treatment of diseases caused by *Trypanosomes* and *Leishmania*. These are conserved proteins that regulate the cell cycle through their kinase activities [18,19]. Here, we described the reactivity of antigen RDK2 with the antibodies in infected sera. RDK2 exhibited a sensitivity of 78% with the confirmed VL cases, and a specificity of 86.67% with non-endemic healthy individuals as negative controls. Immunoblot assay further reveals the binding of RDK2 with the antibodies present in VL sera as compared to control and other diseases.

To reduce the invasiveness of sero-diagnostic tests, we have performed the diagnostic evaluation of RDK2 using a non-invasive biological source, urine. In this study, we found that urine samples exhibited better sensitivity and specificity at 93.8% and 85.7%, respectively, as compared to serum samples. Therefore, according to the results obtained, RDK2 antigen can be applied for non-invasive diagnosis of VL using urine samples. Since the number of relapse cases is high in VL, it is necessary to determine the clinical cure after treatment. Therefore, discrimination of disease before and after VL treatment is important. To serve this purpose, the reactivity of RDK2 was evaluated with follow-up patients, after six months of treatment. The study suggested that urine samples are better than serum in distinguishing active VL from past infections with RDK2, and can thus be explored in the future as a test of cure.

In conclusion, using RDK2 as an antigen in *Leishmania* diagnosis, urine can be a choice of sample over serum for diagnosis as well as for monitoring treatment response in visceral leishmaniasis. However, the study warrants further validation with a large number of samples, and this will be the scope of future studies.

## Figures and Tables

**Figure 1 pathogens-11-00120-f001:**
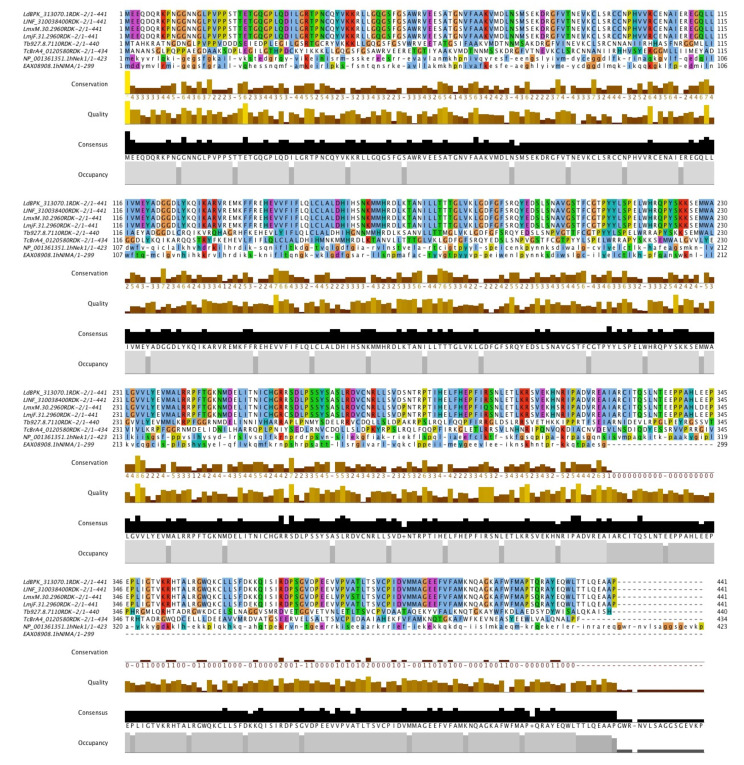
Sequence Comparison of RDK2 ((LdBPK_313070.1RDK-2). Multiple sequence alignments of amino acid sequences analysed using Jalview of recombinant antigen RDK2 comparing its conserved residues with *L. donovani*, *L. infantum*, *L. mexicana*, *L. major*, *T. brucei*, *T. cruzi* and *H. sapiens*.

**Figure 2 pathogens-11-00120-f002:**
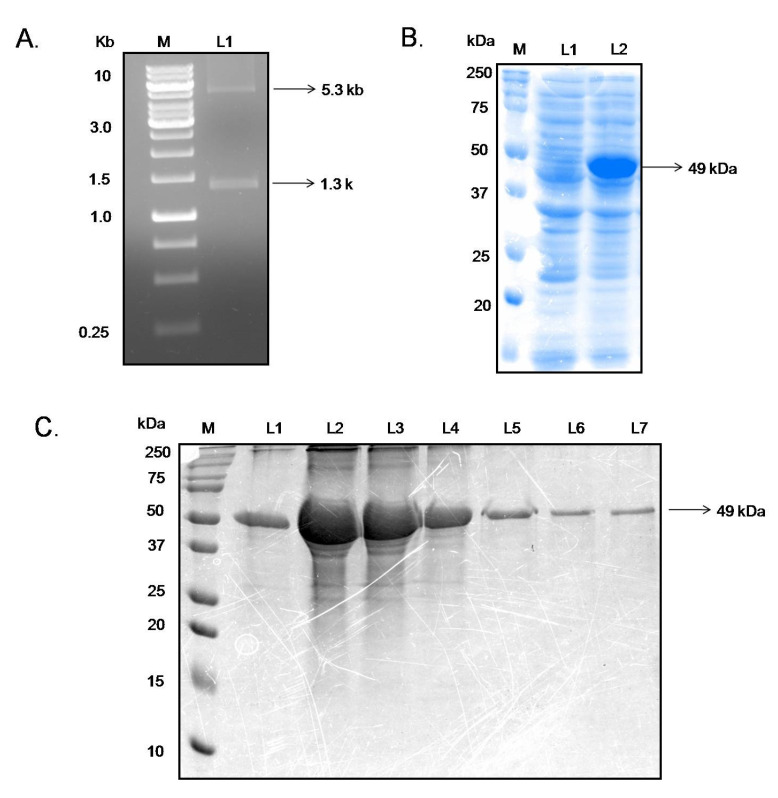
Cloning and expression of purified recombinant proteins RDK2. (**A**) RDK2 was PCR amplified and cloned into pET28a. Marker contains base pair markers, and lane (L1) shows the 5.3 and 1.3 kb PCR product corresponding to the gene length of pET28a and RDK2, respectively. (**B**) Expression of RDK2 upon induction with IPTG. Lane M contains molecular weight markers, lane (L1) represents un-induced lysate, and that of lane (L2) lysate of culture induced with 0.5 mM IPTG, corresponding to a 49-kDa protein. (**C**) SDS-PAGE (12%) analysis of purified RDK2 fractions (20 µL/lane). Molecular weight markers (M) in lane 1 and lanes (L1 to 7) depict recombinant proteins elution fraction.

**Figure 3 pathogens-11-00120-f003:**
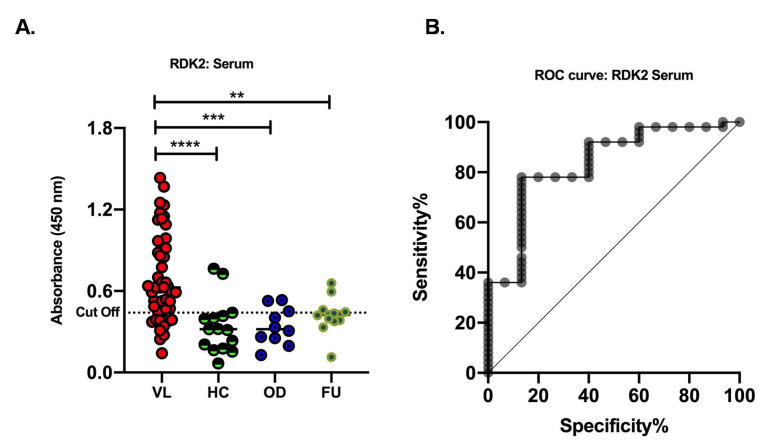
Serum-based ELISAs with recombinant antigens RDK2. (**A**) Sera for this study were collected from active visceral leishmaniasis patients (VL, n  =  50), healthy controls (HC, n = 15) includes (non-endemic healthy controls, n  =  10 and endemic healthy controls n  =  05), other diseases (OD, n  =  10), which included 3 samples each from malaria and viral fever, and 2 samples each from typhoid, tuberculosis patients, and follow-up (FU, n = 13). The dotted horizontal lines represent the cutoff values (0.44) for each ELISA calculated using ROC curves where the highest sensitivity and specificity were obtained. (**, *p* = 0.0022, ***, *p* = 0.0002, ****, *p* < 0.0001). (**B**) ROC curves obtained from ELISAs using RDK2 antigens for detection of antigen-specific antibodies in serum samples.

**Figure 4 pathogens-11-00120-f004:**
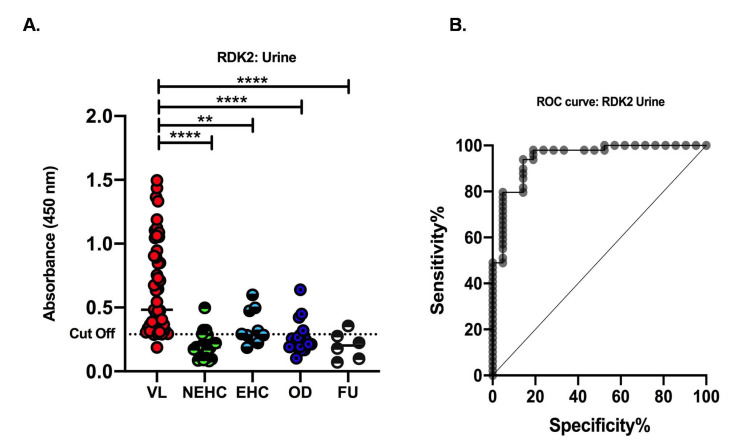
Urine-based ELISA with recombinant antigens RDK2. (**A**) Urine for this study collected from active visceral leishmaniasis patients (VL, n  =  49), non-endemic healthy controls (NEHC, n  =  21), endemic healthy controls (EHC, n  =  10), other diseases (OD, n  =  19), which included 5 samples each from malaria, tuberculosis, and viral fever, and 4 typhoid samples of patients and follow-up (FU, n = 6). The dotted horizontal lines represent the cutoff values (0.29) for each ELISA calculated using ROC curves where the highest sensitivity and specificity were obtained. (**, *p* = 0.0011, ****, *p* < 0.0001). (**B**) ROC curves obtained from ELISAs using RDK2 antigen for detection of antigen-specific antibodies in urine samples.

**Figure 5 pathogens-11-00120-f005:**
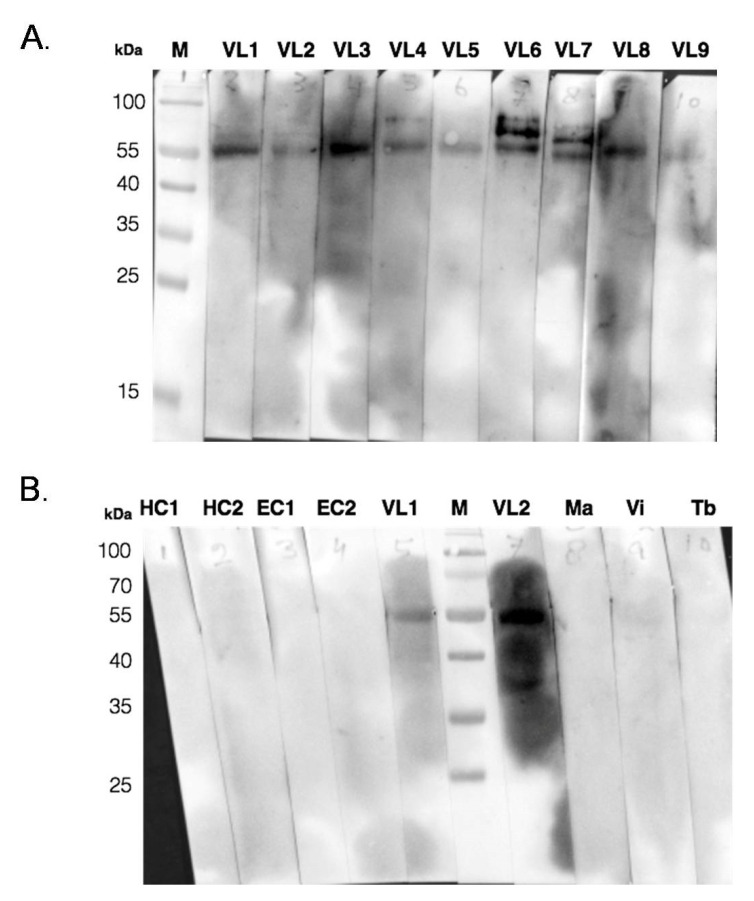
Immunoblot assays using sera against recombinant RDK2. (**A**) Immunoblot assays of recombinant *Leishmania* proteins RDK2 with 9 VL serum samples (VL1 to 9), with protein marker (M) in lane1. (**B**) In another set of experiments, immunoblot assays of RDK2 done with non-endemic healthy controls in lanes 1 and 2 (HC1, HC2), endemic healthy controls in lanes 3 and 4 (EC1, EC2), VL-positive samples in lane 5 and lane 7 (VL1, VL2), protein marker (M) in lane 6 and other diseases in lanes 8 to 10, including malaria (Ma), viral fever (Vi), and tuberculosis (Tb), respectively.

## Data Availability

All the data presented in the study are included in the article. Further enquires can be directed to the corresponding author.

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
