# Peer review of "Revealing a Novel Antigen Repressor of Differentiation Kinase 2 for Diagnosis of Human Visceral Leishmaniasis in India"

_pathogens, 2022, doi:10.3390/pathogens11020120_

Round 1
Reviewer 1 Report
Overview
The authors in the MS entitled “Revealing a Novel Antigen Repressor of Differentiation Kinase 2 2 for Diagnosis of Human Visceral Leishmaniasis in India” cloned and over-expressed a recombinant Leishmania kinase, namely LdRDK2 RDK2 for diagnosis of human VL. Recombinant RDK2 reacted with IgG antibodies in VL patients’ sera with 78% sensitivity and 86.67% specificity as compared to healthy controls and other diseases. The authors showed that recombinant RDK2 could be used as an antigen for non-invasive diagnosis of VL using patients’ urine samples and found 93.8% sensitivity and 85.7% specificity. Overall this MS is well written and useful to the field. The major comment is that before it is published it requires some amendments in the description of the protocols.
Comments
Major
In the section “Cloning, overexpression and purification of L. donovani RDK2” it would be nice if the authors mention whether the cloning in the vector created an in frame tag ie 6-histidine tag.
Also because this paper is quite “technical” it would be nice if in the description of Materials and Methods the authors refer to the amounts. Ie How much volume of E.coli was used and how much lysate was used
The experiment was performed by coating recombinant RDK2 antigen (1mg/ml) in 96-well ELISA plates 137 (Nunc) overnight at 4°C diluted in phosphate buffer (PB).
HRP-conjugated goat anti-human IgG antibody at 1:3,000 dilutions
In the figure legend of figure 2, there is no info on the amounts loaded per lane
Minor
Some typos or English mistakes must be corrected
Ie Multiple sequence alignment should be multiple sequence alignments
Figure legend of figure 1: Please explain what does it mean in the Quality presentation
Author Response
# Response to Reviewer 1
Overview
The authors in the MS entitled “Revealing a Novel Antigen Repressor of Differentiation Kinase 2 2 for Diagnosis of Human Visceral Leishmaniasis in India” cloned and over-expressed a recombinant Leishmania kinase, namely LdRDK2 RDK2 for diagnosis of human VL. Recombinant RDK2 reacted with IgG antibodies in VL patients’ sera with 78% sensitivity and 86.67% specificity as compared to healthy controls and other diseases. The authors showed that recombinant RDK2 could be used as an antigen for non-invasive diagnosis of VL using patients’ urine samples and found 93.8% sensitivity and 85.7% specificity. Overall this MS is well written and useful to the field. The major comment is that before it is published it requires some amendments in the description of the protocols.
Response:
The authors thank the reviewer for his/her appreciation and thank for summarised the work. Below is the point wise response against the comments raised by the reviewer.
MAJOR
Comment: In the section “Cloning, overexpression and purification of L. donovani RDK2” it would be nice if the authors mention whether the cloning in the vector created an in frame tag ie 6-histidine tag.
Response: Thanks for picking this mistake. We have made the changes in the Line number 122 and 123 of Section 2.4. as suggested by the reviewer. The phrase is now changed to “The cloning of RDK2 gene from L. donovani strain AG83 (ATCC PRA-413) was carried out as 6-Histagged protein in pET28a vector”.
Comment: Also because this paper is quite “technical” it would be nice if in the description of Materials and Methods the authors refer to the amounts. Ie How much volume of E.coli was used and how much lysate was used
Response: Thanks to point out this. We have incorporated the suggested description in Line numbers (125, 126, 128, and 129) of section 2.4. as recommended. Now it is “1000 ml of Luria-Bertaini media in the presence of” in Line number 125 and 126, “using a 1000 ml culture of E.Coli” in Line number 128, and “10 ml of bacterial lysis buffer” in Line number 129.
Comment: The experiment was performed by coating recombinant RDK2 antigen (1mg/ml) in 96-well ELISA plates 137 (Nunc) overnight at 4°C diluted in phosphate buffer (PB).
HRP-conjugated goat anti-human IgG antibody at 1:3,000 dilutions
In the figure legend of figure 2, there is no info on the amounts loaded per lane
Response: Thanks to point out this. Information of amount loaded per lane has been added Line number 196 in Figure 2.
MINOR
Comment: Some typos or English mistakes must be corrected.
Response: The authors thank the reviewer for pointing out the typos or English mistakes. We have checked the manuscript thoroughly and errors were rectified in the revised manuscript.
Comment: Ie Multiple sequence alignment should be multiple sequence alignments
Response: Thanks. The correction was made in the Figure 1 legend as suggested by the reviewer (Line number 181).
Comment: Figure legend of figure 1: Please explain what does it mean in the Quality presentation
Response: Alignment Quality is one of the automatically calculated quantitative alignment annotations displayed below the columns of a multiple sequence alignment (and can be used to shade the alignment). It is an ad-hoc measure of the likelihood of observing the mutations (if any) in a particular column of the alignment.
Reviewer 2 Report
The submitted paper is intesting however there are some issue that should be clarified.
minor comments:
- the authors should ameliorate the quality of figures 2 and 5
- introduction: the authors should add some more current references about leishmania and laboratory investigations (doi: 10.1186/s12917-020-2234-9; doi: 10.3390/ani10040557; doi: 10.1007/s00436-020-06845-7.)
- despite the interesting results described in this manuscript, the authors should read throughly their manuscript and check: a) space between words; b) punctuation; c) English of some sentences
Author Response
# Response to Reviewer 2
Comment: The submitted paper is intesting however there are some issue that should be clarified.
Response: The authors thank the reviewer for their appreciation and thank for summarised the work. Below is the point wise response against the comments raised by the reviewer.
Minor Comments:
Comment: The authors should ameliorate the quality of figures 2 and 5.
Response: As suggested by the reviewer, we have replaced the Figure 1 and Figure 2 with high quality pictures in 600 dpi.
Comment: Introduction: the authors should add some more current references about leishmania and laboratory investigations (doi: 10.1186/s12917-020-2234-9; doi: 10.3390/ani10040557; doi: 10.1007/s00436-020-06845-7.)
Response: As recommended, three additional references, References no. 3, 7 and 8) have been added in the revised manuscript.
Comment: despite the interesting results described in this manuscript, the authors should read throughly their manuscript and check: a) space between words; b) punctuation; c) English of some sentences
Response: The authors thank the reviewer for their suggestions about the language, we have checked the manuscript thoroughly and corrections have been made accordingly.
Reviewer 3 Report
Very interesting and valuable study given the incidence of leishmaniasis a same countries and the need to find fast sensitive diagnostic methods capable of allowing a better evaluation of the response to therapy; all this associated with a reduction in implementation costs. The introduction and bibliography are adequate. The experimental design is well described. The discussion consistent with the results. The limit of the work is linked to the low number of samples, wich, as the authors state, will be implemented in the next studies.
Author Response
# Response to Reviewer 3
Comment: Very interesting and valuable study given the incidence of leishmaniasis a same countries and the need to find fast sensitive diagnostic methods capable of allowing a better evaluation of the response to therapy; all this associated with a reduction in implementation costs. The introduction and bibliography are adequate. The experimental design is well described. The discussion consistent with the results. The limit of the work is linked to the low number of samples, wich, as the authors state, will be implemented in the next studies.
Response: The authors thank the reviewer for their appreciation of our work. Author is correct that this study will be further explored in near future with more number of samples.